

# Biological and genetic structure of *Epinephelus costae* (Steindachner, 1878) population in Iskenderun Bay, eastern Mediterranean Sea

Servet Ahmet Doğdu

Vocational School of Maritime Technologies, Underwater Technologies, Iskenderun Technical University, Iskenderun, Hatay, Turkey

## ABSTRACT

This study aimed to status of the Iskenderun population of *Epinephelus costae*, an important fish species in the Mediterranean Sea ecosystem, by providing detailed data on its genetic structure using mtDNA COI gene region and biological parameters as length-weight relationship, age-growth characteristics, von Bertalanffy growth parameters. A total of 325 specimens were studied from Iskenderun Bay in the eastern Mediterranean (Türkiye), ranging in length from 16.7 to 43.5 cm. For genetic studies using the mtDNA COI gene region, 30 samples were selected from different size groups representing the population. Age 2 represented the majority of the population (31.69%). The value of "b" of the length-weight relationship was higher than "3" (3.0839). The results for the von Bertalanffy growth parameters were observed as $L_\alpha = 125.8$ cm; $k = 0.0570$ year $-1$; $t_0 = -2.2410$ year. Fulton's condition factor was observed as 1.4120. The growth performance index was calculated as 4.35. The genetic diversity of the population was found to be $0.0332 \pm 0.0028$. Sixteen haplotypes were observed in the population and haplotype diversity was calculated as 0.9379. Tajima's neutrality test ($-0.9437$), Fu's Fs Statistic ($-0.0340$), Strobeck's S Statistic (0.6620), Achaz's Y test ($-1.8728$), Fu and Li D (1.1570) and F Test (0.4444) were applied to test whether the population is in balance. In conclusion, the biological and genetic analyses carried out within the scope of our study revealed that the Iskenderun population of *E. costae* is in stable condition.

# INTRODUCTION

The Goldblotch grouper *Epinephelus costae* (Steindachner, 1878), formerly known as *E. alexandrinus,* an invalid name now attributed to *E. fasciatus*, is common in the Mediterranean and Eastern Atlantic (*Heemstra & Randall, 1993*). It is a species of high economic value for the Mediterranean. *E. costae* is considered one of the most intriguing species of the Epinephelidae family due to its colour change during life stages (*Tiralongo, Kalogirou & Agostini, 2021*). It is a demersal species, that commonly inhabits sandy, muddy or rocky areas at depths of 4–160 m, but can be distributed up to 300 m depth (*Froese &*

Corresponding author
Servet Ahmet Doğdu,
dogduservet@gmail.com

*Pauly, 2024*). *E. costae*, a hermaphrodite species, is listed as 'Data Deficient' in the IUCN Red List for the world and the Mediterranean Sea (*Zaidi, Derbal & Kara, 2017*; *Francour & Pollard, 2018*).

There are several uses of length-weight ratio (LWR) data in fisheries biology (*Dogdu & Turan, 2024a*). It is especially used in estimating the weight of the fish from its length with length classes, biomass calculations, calculation of stock assessment models, calculation of condition indices and comparison of populations (*Anderson et al., 1983*; *Bilge, Filiz & Yapici, 2017*; *Ergüden, Dogdu & Turan, 2023*; *Dogdu & Turan, 2024b*). LWR data are of critical importance in the assessment of fish stocks, providing fundamental information on the condition and growth of fish (*Le Cren, 1951*; *Ricker, 1975*). This fundamental data facilitates the comparison of species across different populations and habitats (*Le Cren, 1951*; *Ricker, 1975*). It is crucial to gain an understanding of a species' genetic structure before implementing any management and conservation policies, to effectively combat the issue of overfishing (*Dogdu et al., 2022*; *Uyan et al., 2024*). The measurement of genetic diversity and haplotype diversity serves as a crucial indicator for the investigation of mtDNA genetic variation within a population (*Harrison, 1989*; *Jiang et al., 2019*; *Dogdu & Turan, 2021*; *Turan et al., 2024a*). mtDNA COI gene region is a valuable marker for population studies, offering some advantages: its compact size, rapid evolutionary rate, and unique maternal inheritance pattern make it a useful genetic indicator (*Hurst & Jiggins, 2005*; *Xu et al., 2014*; *Turan et al., 2017*; *Turan et al., 2024b*). It can provide accurate information to understand the genetic structure and genetic diversity of the population and determine species stocks to prevent over-exploitation (*Zhao et al., 2019*).

Although there have been some studies on the biology and genetics of *E. costae* to date on; growth (*Wadie et al., 1981*; *Ezzat et al., 1982*; *Bouain, 1986*; *Glamuzina et al., 2003*), length-weight relationship (*Ezzat et al., 1982*; *Can, Basusta & Çekiç, 2002*; *Morey et al., 2003*; *Akyol, Kinacigil & Sevik, 2007*; *Ceyhan, Akyol & Erdem, 2009*; *Tsagarakis et al., 2015*; *Jisr et al., 2018*; *Evagelopoulos et al., 2020*), reproduction (*Bouain & Siau, 1983*), diet (*Diatta, Bouaïn & Capape, 2003*; *Lopez & Orvay, 2005*; *Zaidi, Derbal & Kara, 2017*), colour pattern (*Agostini & Puretti, 2018*; *Tiralongo, Kalogirou & Agostini, 2021*), and genetic diversity (*Elglid et al., 2019*), these are not considered sufficient by the IUCN to understand the stock status of the species in the Mediterranean and worldwide (*Francour & Pollard, 2018*). Therefore, more studies on the biology and genetics of the species are needed.

This study aimed to reveal the status of the Iskenderun population of *Epinephelus costae*, an important fish species in the Mediterranean ecosystem, by providing detailed data on its genetic structure using mtDNA COI gene region and biological parameters as length-weight relationship, age-growth characteristics, von Bertalanffy growth parameters.

## MATERIALS & METHODS

Between January 2024 and April 2024, a total of 325 specimens of *Epinephelus costae* were collected in Iskenderun Bay, Türkiye (Fig. 1). All samples were procured from fish caught by fishermen using a commercial trammel net (Fig. 2). The reason for choosing these sampling areas is that they cover the natural habitats of the species and the intensive fishing activities

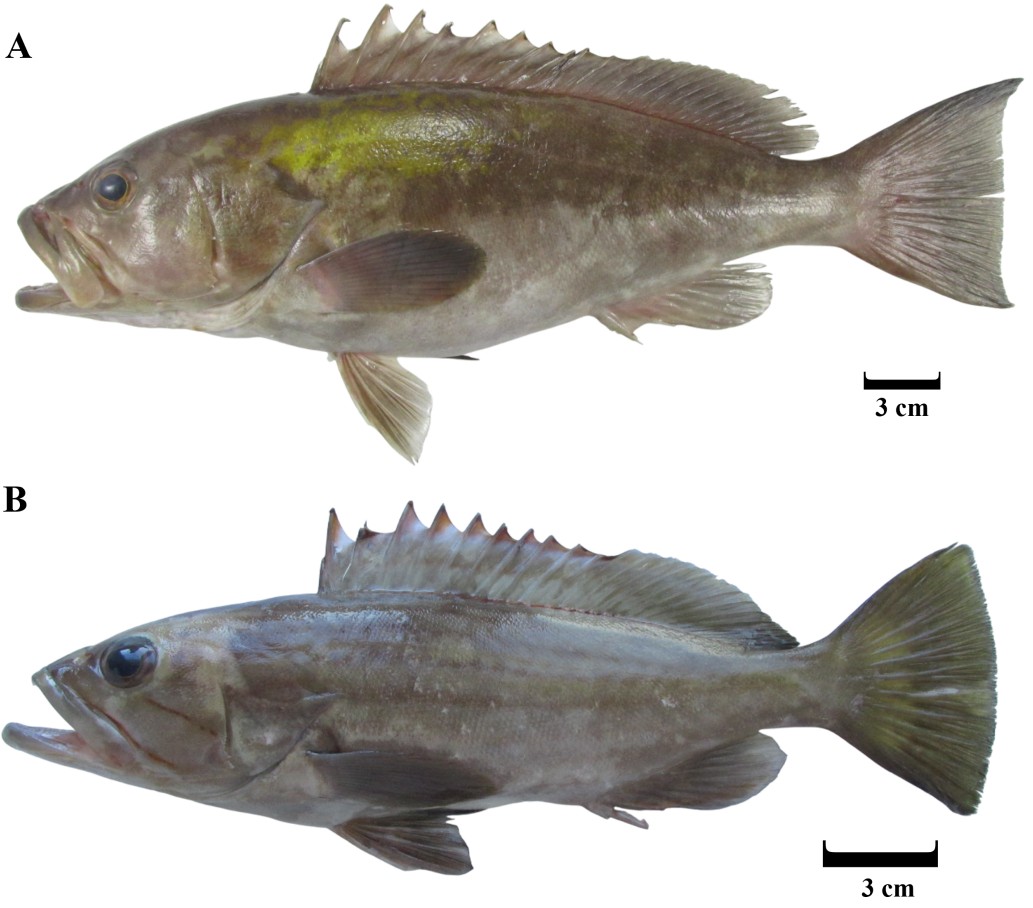

**Figure 1** Adult (A) and juvenile (B) *E. costae* specimens captured during the study. (A) 39.3 cm, (B) 24.8 cm.

carried out in Iskenderun Bay are carried out at these points. After collection, the samples were transported to the laboratory without breaking the cold chain and stored at −21 °C until morphologic and genetic analyses. In the laboratory, each fish was subjected to a comprehensive assessment, during which its total length (TL) was measured in centimeters (cm) and its weight (W) was recorded with an accuracy of 0.01 grams. Since dead fish material was used in the study, it does not require Local Ethics Committee Approval since experimental animals were not used.

Age identification was performed by analyzing the fish scales. The scales were relatively deep; therefore, with the help of forceps and a spear-tipped dissecting needle, scale samples were taken from the areas close to the dorsal fin, starting from the bottom of the pectoral fin towards the tail. Scales of the same size were generally preferred and 20 scales were taken from each fish sample. Three separate readers independently validated the results on three occasions, using a Nikon SMZ–U stereo zoom microscope at 10× magnification to determine age.

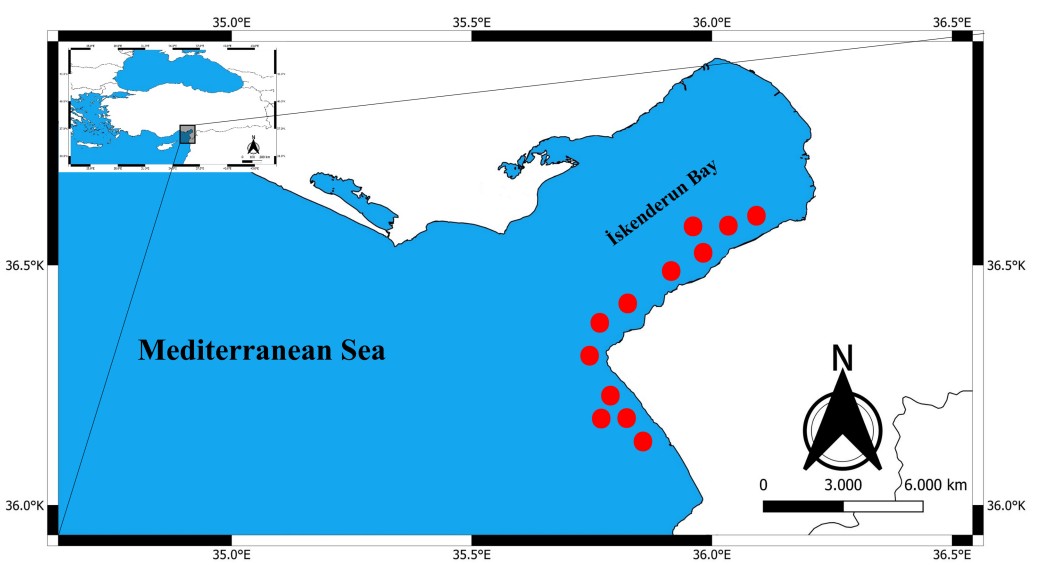

**Figure 2  Commercial trawling areas (red dots) where *E. costae* specimens were caught.**

The length–weight relationship was derived using the equation proposed by *Ricker (1975)*:

$$W = a \times TL^b.$$

The abbreviations used in the formulas are: $W$ for weight (g), $TL$ for total length (cm), $b$ as the length–weight exponent, and $a$ as a constant. Length–weight relationships were calculated for all samples. To assess differences in regression slopes and average lengths all samples, we applied a $t$-test (*Zar, 1999*). All statistical computations were conducted in SPSS version 22.0.

The parameters $a$ and $b$ were calculated through the implementation of least-squares regression, with the coefficient of determination ($R^2$) serving as a metric for evaluating the fit of the model to the data. The significance of the regression analysis was determined using an analysis of ANOVA, according to *Zar (1999)*. A Student's $t$-test was used to ascertain whether the b values obtained from linear regressions were significantly different from the null hypothesis of isometric growth (H0: $b = 3$) using the equation proposed by *Sokal & Rohlf (1987)*.

$$t_s = (b-3)/sb.$$

In this equation, $ts$ denotes the $t$-test value, $b$ signifies the slope, and $sb$ represents the standard error of the slope ($b$). The statistical analyses were conducted using Microsoft Excel 2016 (Microsoft, Redmond, WA, USA) and SPSS Statistics 19.0 (SPSS Inc., Chicago, IL, USA).

Growth parameters were estimated using the von Bertalanffy growth model as described by *Von Bertalanffy (1938)*;

$$L_t = L_\infty(1 - e^{-k(t-t0)}).$$

The model parameters are defined as: $L_t$ for total length at age $t$, $K$ as the growth rate constant, $L_\infty$ for the theoretical maximum length, and $t_0$ for the hypothetical age corresponding to zero length.

Using *Pauly & Munro*'s (*1984*) formula, we calculated the growth performance index ($\Phi'$), a length-based metric for evaluating growth performance.

$$\emptyset = log_{10}K + 2log_{10}L_\infty.$$

Genetic analyses were carried out on thirty individuals covering a range of size classes within the population. Genomic DNA was isolated from muscle tissue using a modified phenol–chloroform–isoamyl alcohol extraction protocol (*Sambrook, Fritsch & Maniatis, 1989*). The mtDNA COI gene was amplified through polymerase chain reaction (PCR) with universal primers (COI F: 5′-TCA ACC AAC CAC AAA GAC ATT GGC AC-'3′, COI-R: 5′-ACT TCA GGG TGA CCG AAG AAT CAG AA-'3) (*Ward et al., 2005*). The PCR was carried out in a total volume of 50 µl. The volume comprised 0.5 µM of each primer, 0.2 mM of dNTP and 1 U of Taq DNA polymerase. The PCR buffer contained 20 mM of Tris–HCl (pH 8.0), 1.5 mM of MgCl$_2$, 15 mM of KCl, and 1.0 µl of template DNA. The denaturation stage was initiated at 95 °C for 45 s, followed by a 40-second extension at 59 °C and a 60-second extension at 72 °C. This sequence was repeated for a total of 30 cycles, culminating in a final extension step at 75 °C for a duration of 5 min. The PCR products were visualized using electrophoresis on 1.5% agarose gel. The DNA sequencing was conducted using a chain termination method, as originally described by *Sanger, Nicklen & Coulson (1977)*, utilizing the BigDye Cycle Sequencing Kit V3.1 and the ABI 3130 XL genetic analyzer. The initial partial mtDNA sequence alignments were performed using Clustal W (*Thompson, Higgins & Gibson, 1994*) in MEGAX software (*Kumar et al., 2018*), and the final alignment was completed manually with BioEdit (*Hall, Biosciences & Carlsbad, 2011*). The calculations of haplotype diversity, genetic diversity (*Nei, 1987*) and the mean number of pair-wise differences (*Tajima, 1983*) were conducted utilizing the Arlequin (*Schneider, Roessli & Excoffier, 2000*) program. The haplotypes of the sequences were determined using the DnaSP 6 program (*Rozas et al., 2017*).

# RESULTS

A total of 325 *E. costae* samples were analysed in this study. The total length range was 16.7–43.5 cm and the total weight ranged from 51.75–1,020.90 g. The mean total length and total weight of all specimens were observed as 28.8 ± 6.5 cm and 337.40 ± 242.52 g, respectively. Fish scales were analysed and the maximum age was determined as 5 years and 0 age was not observed (Table 1).

The length-weight relationship of *E. costae* was calculated as $W = 0.009 \times L^{3.0839}$ ($R^2 = 0.9941$) (Fig. 3). The '$b$' value obtained by least squares regression is higher than '3', indicating that this population exhibits positive allometric growth. The value of $b = 3.0839$ obtained in the regression analysis of the height-weight relationship showed a statistically significant deviation from the theoretical value of $b = 3$, which represents isometric growth ($t = 3.93 > t_{0-005} = 1.96$; N > 300). This suggests that in the studied population, body

**Table 1  Mean total length and weight values for each age group of *E. costae* (SD: Standard deviation).**

| Age | N | % | TL min–max (cm) | Mean TL ± SD (cm) | W min–max (g) | Mean W ± SD |
|---|---|---|---|---|---|---|
| 1 | 98 | 30.15 | 16.7–24.2 | 21.8 ± 1.5 | 51.75–176.24 | 125.91 ± 24.05 |
| 2 | 103 | 31.69 | 26.6–1.9 | 26.6 ± 1.9 | 154.56–410.25 | 223.31 ± 58.15 |
| 3 | 68 | 20.93 | 30.4–37.3 | 33.4 ± 1.4 | 330.54–636.25 | 454.40 ± 67.39 |
| 4 | 52 | 16 | 36.1–42.5 | 39.1 ± 2.0 | 572.58–950.08 | 760.10 ± 128.78 |
| 5 | 4 | 1.23 | 43.1–43.5 | 43.3 ± 0.2 | 962.90–1,020.9 | 972.63 ± 37.46 |
| Total | 325 | 100 | 16.7–43.5 | 28.8 ± 6.5 | 51.75–1,020.90 | 337.40 ± 242.52 |

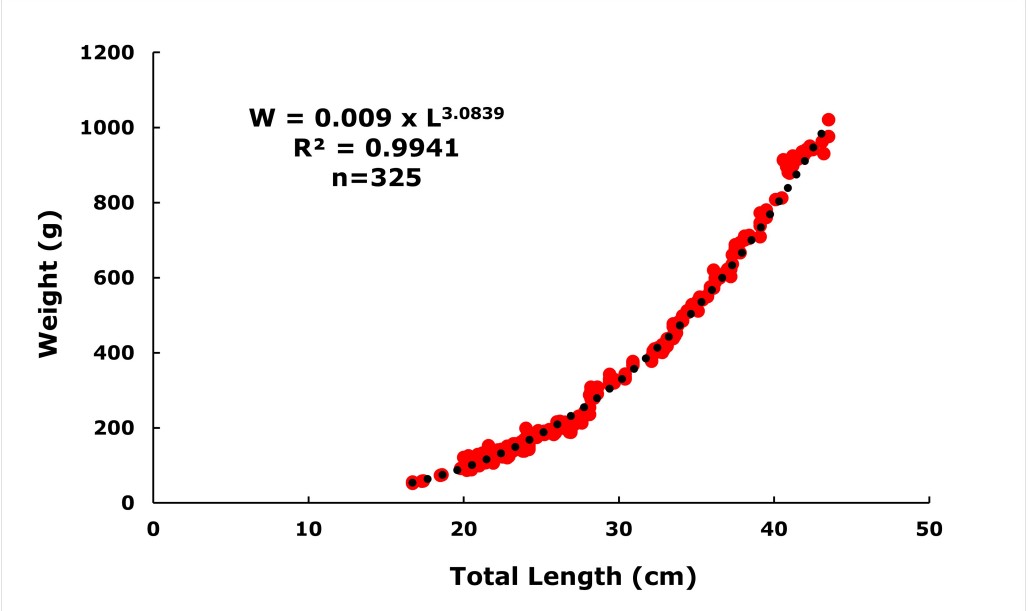

**Figure 3  Length-weight relationships of *E. costae* specimens.**

mass increased proportionally more with the increase in length of individuals, and thus they exhibited a positive allometric growth. Such a growth pattern can often be attributed to the abundance of food resources, favorable environmental conditions and the genetic growth capacity of the species.

Age classes in the *E. costae* population spanned 1–5 years, with two-year-olds representing the largest cohort (31.69%). Detailed descriptive statistics are presented in Table 1, and the full age frequency distribution is illustrated in Fig. 4.

Growth parameters were obtained by the data to the von Bertalanffy equation (Fig. 5). The von Bertalanffy growth parameter results were obtained as $L = 125.8$ cm; $k = 0.0570$ year$^{-1}$; $t_0 = -2.2410$ year.

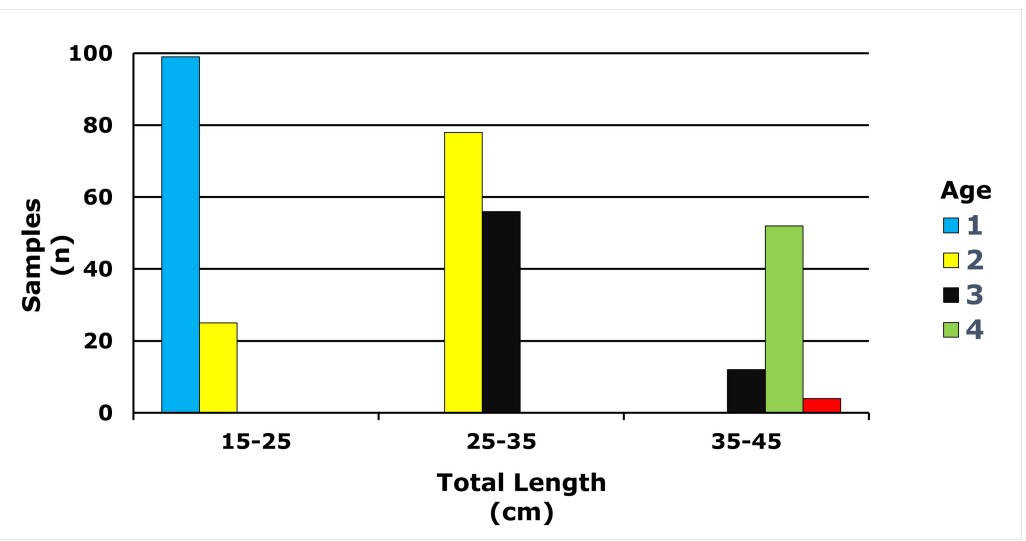

**Figure 4** Age-length distribution of *E. costae* population.

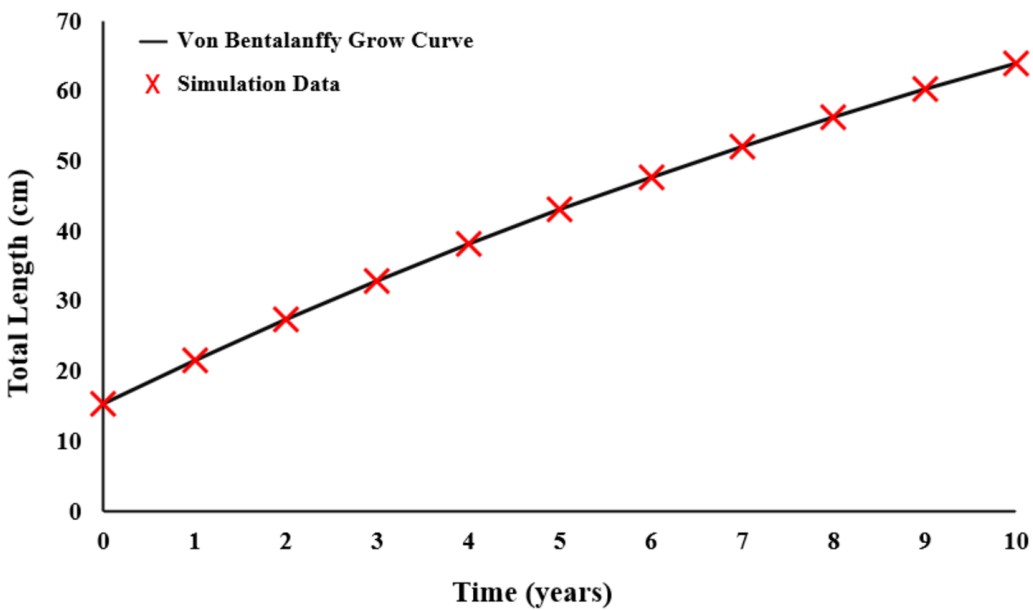

**Figure 5** Von Bertalanffy growth curve of *E. costae*.

Fulton's condition factor (K) was calculated using the weight (Fig. 6). The condition factor was observed at 1.4120 for all specimens. A *K* value greater than 1 is an indication that the population is well-fed and growing well.

The growth performance index (Ø) was calculated as 4.35. This value indicates that the species generally tends to grow faster and has a potentially long life span. *E. costae* stand

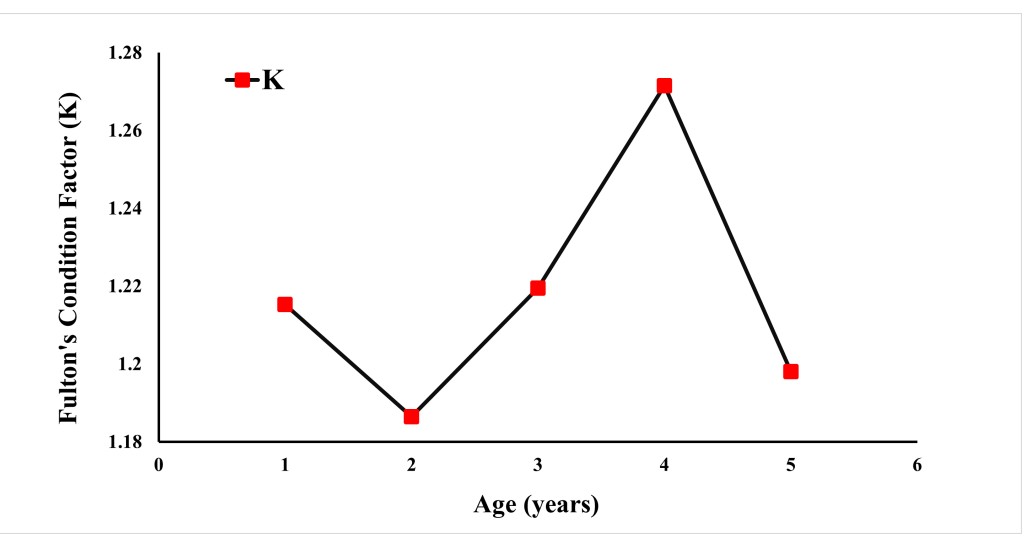

**Figure 6** Fulton condition factor of *E. costae* by age.

out as an economically and ecologically valuable species with its high growth rates and large size potential.

A genetic study of 30 samples representing 325 *E. costae* individuals captured from Iskenderun Bay was carried out. After alignment, the mtDNA COI dataset revealed 120 variable, 18 singleton and 530 conservative nucleotide sites, 102 of which were parsimony informative over 650 bp sequences. The average nucleotide variation of the sequences obtained was 31.3% for thymine (T), 26.1% for cytosine (C), 24.3% for adenine (A) and 18.3% for guanine (G). The best model Kimura 2 (K2) was provided by the MEGAX software (*Kimura, 1980*; *Kumar et al., 2018*). The transition/transversion ratio (R) was calculated as 1.35. The sequences of the samples used in the study were registered in the NCBI database with accession numbers PQ814215–PQ814244.

Overall genetic diversity of sequences was observed at 0.0332 ± 0.0028 with standard error. Pairwise distance analyses between all sequences are shown in Appendix 1. This analysis of 30 nucleotide sequences was performed using the K2 parameter model (*Kimura, 1980*). For each sequence, all ambiguous positions were removed to obtain the final dataset. The analyses showed that 16 haplotypes were observed in the population and haplotype diversity was 0.9379.

To test whether the population was in balance, the neutrality test of *Tajima (1989)*, Fu's Fs Statistic (*Fu, 1997*), Strobeck's S Statistic (*Strobeck, 1987*), Fu and Li D and F Test (*Fu & Li, 1993*) and Achaz's Y test (*Achaz, 2008*) were applied. The Tajima neutrality test (D) value was found as −0.9437. Fu's Fs test value was found as −0.0340. Strobeck's S Statistic was found as 0.6620. Achaz's Y test was found as −1.8728. Fu and Li's D and F tests were found as 1.1570 and 0.4444, respectively (Fig. 7). Tajima's D and Achaz Y tests, taken together, indicate the possibility of population expansion. This expansion is also confirmed by Strobeck's S test. However, since Fu's Fs is a very weak negative value, this expansion is not strongly supported. Furthermore, Fu's and Li's tests are positive but not

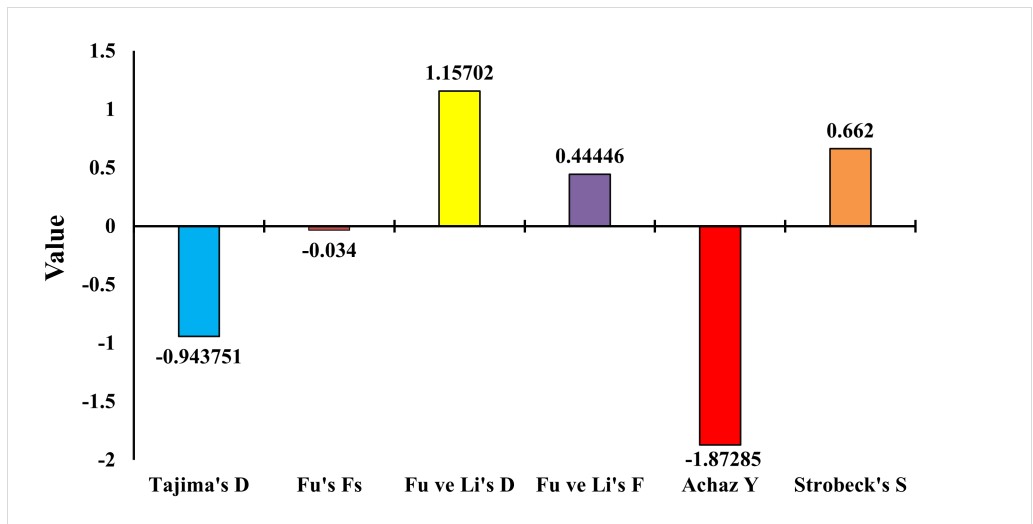

**Figure 7** Statistical analysis results of *E. costae* Iskenderun population.

significant and therefore not considered. Additional tests or analyses on different loci may be recommended to obtain stronger results.

## DISCUSSION

Understanding the genetic structure and growth parameters of fish populations is essential for effective stock assessment and sustainable fisheries management. This study contributes to closing knowledge gaps for *Epinephelus costae* in the Eastern Mediterranean, particularly in Iskenderun Bay, by providing new data on growth dynamics and genetic variability.

In this study, the coefficient of determination ($R^2$) for the 325 samples of *Epinephelus costae* analyses in the study was 0.9941. The length-weight relationship (LWR) showed a strong correlation between the length and weight of the samples. *Can, Basusta & Çekiç (2002)* studied the $R^2$ of *E. costae* for 0.93 in Iskenderun Bay, Türkiye. *Morey et al. (2003)* reported the $R^2$ of *E. costae* for 0.988 on the Mediterranean coast of Spain. *Akyol, Kinacigil & Sevik (2007)* determined the $R^2$ value of *E. costae* as 0.966 on the Gökova Bay of Türkiye. *Ceyhan, Akyol & Erdem (2009)* studied the $R^2$ of *E. costae* for 0.942 in Gökova Bay of Türkiye. *Kouassi, N'da & Soro (2010)* reported the $R^2$ of *E. alexandrinus* for 0.813 in Atlantic Ocean of Ivery Coast. *Tsagarakis et al. (2015)* determined the $R^2$ value of *E. costae* as 0.986 on the Antalya Bay of Türkiye. *Çete (2017)* studied the $R^2$ of *E. costae* for 0.914 on the eastern Mediterranean coast of Türkiye. *Jisr et al. (2018)* determined the $R^2$ of *E. costae* for 0.898 in Lebanon. *Evagelopoulos et al. (2020)* reported the $R^2$ of *E. costae* for 0.970 in the Ionian Sea of Greece. It was observed that the $R^2$ values found in studies on *E. costae* and other Epinephelus species were close to each other (Table 2). Compared to other regional studies, the $R^2$ values were largely consistent, though some variation is likely due to factors such as sex, reproductive status, feeding and preservation techniques, which were not controlled in this study (*Karachle & Stergiou, 2008*; *Dogdu & Turan, 2024a*; *Dogdu & Turan, 2024b*; *Uyan et al., 2024*).

**Table 2** Growth parameters ($L_\infty$, $k$ and $t_0$), growth performance index (Ø), condition factor (K) and length-weight relationships parameters (a, b and $R^2$) for present and previous studies on *E. costae* and other *Epinephelus* species distrubited in Mediterranean Sea. Note: *Epinephelus alexandrines* is a synonym of *E. costae*.

| References | Species | N | Country | $L_\infty$ (cm) | k(/y) | $t_0$ | Ø | K | a | b | $R^2$ |
|---|---|---|---|---|---|---|---|---|---|---|---|
| *Ezzat et al. (1982)* | *E. alexandrinus* | 360 | Egypt | – | – | – | – | 1.04–1.23 | 0.024 | 2.776 | – |
| | *E. alexandrinus* | – | | 121.59 | 0.041 | −2.059 | – | – | – | – | – |
| *Bouain (1986)* | *E. aeneus* | – | Tunisie | 190.8 | 0.040 | −1.091 | – | – | – | – | – |
| | *E. guaza* | – | | 197.79 | 0.025 | −1.459 | – | – | – | – | – |
| | *E. caninus* | – | | 117.13 | 0.048 | −1.288 | – | – | – | – | – |
| *Bouchereau, Body & Chauvet (1999)* | *E. marginatus* | – | France | 135.91 | 0.079 | −0.803 | – | – | 0.125 | 2.60 | 0.948 |
| *Can, Basusta & Çekiç (2002)* | *E. costae* | 53 | Türkiye | | | | | | 0.088 | 2.391 | 0.93 |
| *Morey et al. (2003)* | *E. costae* | 16 | Spain | – | – | – | – | – | 0.013 | 2.967 | 0.988 |
| | *E. marginatus* | 222 | | – | – | – | – | – | 0.010 | 3.121 | 0.995 |
| *Akyol, Kinacigil & Sevik (2007)* | *E. aeneus* | 125 | Türkiye | – | – | – | – | – | 0.017 | 2.855 | 0.942 |
| | *E. costae* | 59 | Türkiye | – | – | – | – | – | 0.026 | 2.736 | 0.966 |
| *Sangun, Akamca & Akar (2007)* | *E. aeneus* | 24 | Türkiye | – | – | – | – | – | 0.012 | 2.987 | 0.990 |
| | *E. marginatus* | 48 | Türkiye | – | – | – | – | – | 0.011 | 3.065 | 0.910 |
| *Renones et al. (2007)* | *E. marginatus* | 349 | Spain | 95.6 | 0.087 | −1.12 | – | – | – | – | – |
| *Ceyhan, Akyol & Erdem (2009)* | *E. aeneus* | 36 | Türkiye | – | – | – | – | – | 0.009 | 3.043 | 0.952 |
| | *E. costae* | 365 | Türkiye | – | – | – | – | – | 0.017 | 2.885 | 0.942 |
| *Kouassi, N'da & Soro (2010)* | *E. alexandrinus* | 176 | Ivory | – | – | – | – | – | 0.02 | 2.88 | 0.813 |
| *Edelist et al. (2011)* | *E. aeneus* | 34 | Israel | – | – | – | – | – | 0.013 | 2.96 | 0.996 |
| *Özbek, Kebapçioglu & Çardak (2013)* | *E. aeneus* | 350 | Türkiye | – | – | – | – | – | 0.040 | 2.657 | 0.958 |
| *Özvarol & Gökoglu (2015)* | *H. haifensis* | 96 | Türkiye | – | – | – | – | – | 0.009 | 3.142 | 0.996 |
| | *M. rubra* | 74 | Türkiye | – | – | – | – | – | 0.008 | 3.065 | 0.989 |
| | *E. aeneus* | 522 | Türkiye | – | – | – | – | – | 0.009 | 3.059 | 0.989 |
| *Tsagarakis et al. (2015)* | *E. marginatus* | 78 | Türkiye | – | – | – | – | – | 0.001 | 3.108 | 0.978 |
| | *E. costae* | 62 | Türkiye | – | – | – | – | – | 0.009 | 3.051 | 0.986 |
| *Saleh & Ali (2017)* | *E. guaza* | 207 | Libya | – | – | – | – | 1.23–1.88 | 0.029 | 2.969 | 0.99 |
| *Çete (2017)* | *E. aeneus* | 80 | Türkiye | 130.93 | 0.098 | −2.926 | 4.587 | 1.24 | 0.016 | 3.114 | 0.961 |
| | *E. costae* | 19 | Türkiye | 71.29 | 0.175 | −1.110 | 4.071 | 0.92 | 0.003 | 3.310 | 0.914 |
| *Jisr et al. (2018)* | *E. costae* | 13 | Lebanon | – | – | – | – | 1.002 | 0.075 | 2.462 | 0.898 |
| | *E. marginatus* | 13 | | – | – | – | – | 0.995 | 0.019 | 2.906 | 0.897 |
| *Evagelopoulos et al. (2020)* | *E. costae* | 9 | Greece | – | – | – | – | – | 0.013 | 2.942 | 0.970 |
| *El-Aiatt (2021)* | *E. aeneus* | 697 | Egypt | – | – | – | – | 1.1–1.5 | 0.008 | 3.114 | 0.961 |
| *Mehanna & Farouk (2021)* | *E. aeneus* | 98 | Egypt | – | – | – | – | – | 0.055 | 2.724 | 0.97 |
| This study | *E. costae* | 325 | Türkiye | 125.89 | 0.057 | −2.241 | 4.35 | 1.412 | 0.009 | 3.083 | 0.994 |

The *b* value from the LWR equations was found to be 3.083, which is higher than most of the values obtained from previous studies on the same species and genus (Table 2). *Ezzat et al. (1982)* reported that *b* values of *E. alexandrinus* for 2.776 (negative allometry)

in the Mediterranean Sea of Egypt. *Can, Basusta & Çekiç (2002)* reported that *b* values of *E. costae* for 2.391 (negative allometry) in Iskenderun Bay, Türkiye. *Morey et al. (2003)* determined the *b* of *E. costae* for 2.967 (negative allometry) in the Mediterranean coast of Spain. *Akyol, Kinacigil & Sevik (2007)* reported the *b* value of *E. costae* as 2.736 (negative allometry) on the Gökova Bay of Türkiye. *Ceyhan, Akyol & Erdem (2009)* studied the *b* of *E. costae* for 2.885 (negative allometry) in Gökova Bay of Türkiye. *Kouassi, N'da & Soro (2010)* reported the *b* of *E. costae* for 2.88 (negative allometry) in the Atlantic Ocean of Ivery Coast. *Tsagarakis et al. (2015)* determined the *b* value of *E. costae* as 3.051 (positive allometry) on the Antalya Bay of Türkiye. *Çete (2017)* studied the *b* of *E. costae* for 3.310 (positive allometry) in the eastern Mediterranean coast of Türkiye. *Jisr et al. (2018)* determined the *b* of *E. costae* for 2.462 (negative allometry) in Lebanon. *Evagelopoulos et al. (2020)* reported the *b* of *E. costae* for 2.942 (negative allometry) in the Ionian Sea of Greece. A *b* value close to three indicates that the body weight of the fish increases in parallel with the growth in length. Our study showed positive allometric ($p < 0.05$) growth. However, while most studies on the species showed negative allometric growth, positive allometric growth was observed only in studies by *Tsagarakis et al. (2015)* and *Çete (2017)*. These differences in *b* values may be the result of many reasons such as environmental factors, biological parameters and differences in sampling type, size ranges of samples, number of individuals collected and collection time (*Ricker, 1975*; *Turan et al., 2021*; *Ergüden, Dogdu & Turan, 2023*; *Yaglioglu et al., 2025*).

Evaluating fish stock status depends on core growth metrics—namely the asymptotic length ($L_\infty$), the growth coefficient (k) and the hypothetical age at which length equals zero ($t_0$), all of which supported diverse modeling approaches and illuminate population growth dynamics and age structure. This information is of crucial importance for the effective management and conservation of fisheries (*Ergüden & Dogdu, 2020*; *Turan et al., 2021*). These statistics are of significant value in facilitating a comprehensive comparison of fish growth, both between different species and within the same species across various periods and geographic locations. These tools are of paramount importance in comprehending the dynamics of fish populations and facilitating informed decision-making in the domains of fisheries management and conservation (*Maunder & Punt, 2013*). The present study observed length (L), growth rate (k), and age at zero length ($t_0$) of *E. costae* as 125.89 cm, 0.057 and $-2.241$, respectively. A comparative analysis was conducted on the current values of L, k, and $t_0$ with those previously reported in the literature concerning *E. costae* and other *Epinephelus* species (Table 2). The asymptotic length (L) observed in the present study and previous studies is relatively similar, but highly divergent with some studies. These similarities and differences indicate that Iskenderun Bay has favourable environmental conditions for *E. costae* and promotes consistent growth patterns for this species. The results, when compared with the literature, reinforce the importance of localized growth assessments to avoid generalized management strategies. Differences in growth across regions highlight the necessity of site-specific data for accurate stock modelling. Observed size range (16.7–43.5 cm), it is likely that the sampled individuals represent primarily juvenile or subadult stages. This may influence the estimation of growth parameters such
as asymptotic length and should be considered when interpreting population dynamics and management implications.

The genetic structure of the *E. costae* in the Eastern Mediterranean Sea population. In our study, 30 DNA sequences of various lengths representing the population were analysed. The 650 bc long sequences analysed using the COI gene region have 120 variables, 18 singletons, 530 conservative nucleotide sites and 102 parsimony informative sites. The average nucleotide diversity of these sequences was 31.3% for thymine (T), 26.1% for cytosine (C), 24.3% for adenine (A) and 18.3% for guanine (G). The estimated Transition/Transformation bias (R) was found to be 1.35. *Chen et al. (2016)* used mtDNA control region in their study on the genetic structure of *Epinephelus akaara* species and defined the gene length as 331 bp and found 265 bp of these sequences as a conserved region. *Mohammed-Geba (2015)* used the COI gene in his study to investigate the population structure of *Epinephelus akaara* species and found the length of the gene region as 633 bp and 17 bp as a variable region. *Galal-Khallaf et al. (2019)* performed DNA barcoding of sixteen Epinephelus species distributed along the Red Sea coast of Egypt using COI and 12s rRNA gene regions and found 195 parsimony-informative regions among species in COI gene nucleotide sequences. *Fadli, Muchlisin & Siti-Azizah (2021)* found the nucleotide composition as $A = 24.15\%$, $T = 29.56\%$, $C = 28.14\%$ and $G = 18.14\%$ in their DNA barcoding of twenty-six Epinephelus species using the COI gene region on the Indonesian coast. They also detected 403 conserved sites, 236 variable sites, 230 parsimony informative and six singletons in 639 bc sequences. *Fadli et al. (2023)* investigated the DNA barcoding and genetic diversity of six Epinephelus species in Indonesia using COI gene region and found the nucleotide composition $= 24.95\%$, $T = 29.47\%$, $C = 27.79\%$, and $G = 17.79\%$. In our study, the thymine ratio was slightly higher and the guanine ratio was relatively low compared to other species, and the high ratio of conserved region bp compared to other studies may be an indication that the Iskenderun population is genetically more stable. However, the low levels of variable and parsimony informative regions compared to other studies suggest that the genetic diversity of *E. costae* may be lower than in some Indonesian populations. In terms of gene length, the *E. costae* population has a similar gene length to other Epinephelus species.

In our study, the genetic diversity of the population was determined as $0.0332 \pm 0.0028$. *Chen et al. (2016)* used the D-Loop gene region in their study of the genetic structure of *Epinephelus akaara* species and defined the genetic diversity as 0.0271. *Mohammed-Geba (2015)* used the COI gene in his study to investigate the population structure of *Epinephelus akaara* species and found the genetic diversity as 0.0287. *Elglid et al. (2019)* in a population genetics study of *E. costae* from the Zarzis and Tadjourah coasts of Libya using Cyt b and microsatellites genes, found low genetic diversity in both populations (0.001, 0.002) in Cyt b gene region analyses. *Galal-Khallaf et al. (2019)* performed DNA barcoding of sixteen Epinephelus species distributed along the Red Sea coast of Egypt using COI and 12s rRNA gene regions and found genetic diversity as 0.0381 in the COI gene. *Fadli, Muchlisin & Siti-Azizah (2021)* found the genetic diversity as 0.0415 in their DNA barcoding of twenty-six Epinephelus species using the COI gene region on the Indonesian coast. *Fadli et al. (2023)* investigated the DNA barcoding and genetic diversity of six Epinephelus species

in Indonesia using the COI gene and found a genetic diversity of 0.0452. The genetic diversity value obtained in our is in the middle level compared to other Epinephelus species in the literature. Although the genetic diversity value of the Iskenderun population is not as high as the Indonesian population, it is seen that it has higher genetic diversity than the populations of Epinephelus species in China and the Red Sea. Although the genetic diversity value of the Iskenderun population is at a positive level, it is seen that genetic bottlenecks may occur as in China if the hunting pressure on the population continues.

The haplotype diversity value (0.9379) in our study is one of the highest values compared to other Epinephelus species in the literature. *Elglid et al. (2019)* in their population genetics study of *E. costae* in the Zarzis and Tadjourah coasts of Libya, determined haplotype diversity in both populations as 0.833 ± 0.054 and 0.769 ± 0.120 in Cyt b gene region analyses, respectively. *Galal-Khallaf et al. (2019)* found the haplotype diversity value as 0.879 in their study on Epinephelus species on the Red Sea coast of Egypt. *Fadli, Muchlisin & Siti-Azizah (2021)* found the haplotype disparity value as 0.911 in their study on Epinephelus species in Indonesia. *Fadli et al. (2023)* reported a haplotype disparity value of 0.925 in their study on Epinephelus species in Indonesia. *Vu & Nguyen (2024)* reported the haplotype disparity value as 0.903 in their study on Epinephelus species on the coasts of the China Sea. *Loh et al. (2024)* reported the haplotype disparity value as 0.872 in their study on *Epinephelus flavocaeruleus* species on the Malaysian Coast. It has a similar level of diversity to populations in Indonesia (*Fadli, Muchlisin & Siti-Azizah, 2021*; *Fadli et al., 2023*), but higher than populations in the Red Sea (*Galal-Khallaf et al., 2019*), South China Sea (*Vu & Nguyen, 2024*) and Malaysia (*Loh et al., 2024*). The high haplotype diversity obtained also supports our genetic diversity data. This indicates that the *E. costae* population in Iskenderun Bay is at a genetically stable condition. However, it should not be forgotten that factors such as intensive fishing, habitat destruction and climate change, which are likely to be effective on the population, may reduce this haplotype diversity.

In our study for the *E. costae* Iskenderun population, Tajima's neutrality test (D) value was observed at −0.9437, Fu's Fs test value was observed at −0.034, Strobeck's S Statistic was observed at 0.6620, Achaz's Y test was observed −1.87285 and Fu and Li's D and F tests were 1.1570 and 0.4444, respectively. *Galal-Khallaf et al. (2019)* found the D value of Tajima as −0.85 in their study using the COI gene region on the Red Sea coast of Egypt. In addition, determined the Y value of Achaz as −1.75 in his study. *Fadli, Muchlisin & Siti-Azizah (2021)* found Tajima's D value as −1.45 in their study for twenty-six Epinephelus species using the COI gene region on the Indonesian coast. In addition, Fu and Li's D and F tests were found to be −0.98 and −0.54, respectively. *Fadli et al. (2023)* found Fu's Fs value as −3.21 in their study for six Epinephelus species using the COI gene region on the Indonesian coast. They also determined Strobeck's S Statistic as 0.58 in his study. *Vu & Nguyen (2024)* reported Fu's Fs value as −1.89 for Epinephelus species in the South China Sea. The results obtained show that although Tajima's D, Strobeck's S and Achaz's Y tests taken together show that the population tends to expand, this expansion is not strong because Fu's Fs, Fu and Li's D and F tests are weak.

## CONCLUSIONS

The biological analyses results show that positive allometric growth and relatively high asymptotic length indicates that larger, mature individuals play a critical role in the population's biomass and reproductive output (*Ricker, 1975*; *Froese, 2006*). Therefore, management strategies that protect these individuals such as minimum and maximum size limits or slot limits may enhance recruitment success and overall stock resilience (*Tsikliras & Stergiou, 2014*). Genetic findings reveal a population with stable diversity but susceptible to degradation under continued fishing pressure or habitat disturbance. To safeguard long-term viability, it is essential to monitor genetic variation regularly and mitigate factors that could reduce diversity, such as overfishing and environmental degradation (*Allendorf et al., 2008*; *Hauser & Carvalho, 2008*).

The study supports the need for site-specific management strategies tailored to the biological and genetic characteristics of local E. costae populations. These may include seasonal fishing closures during key reproductive periods, the establishment of marine protected areas, and the integration of molecular and growth data into stock assessment models (*Hilborn, Punt & Orensanz, 2004*; *FAO, 2020*).

In conclusion, this study presents comprehensive biological and genetic data for Epinephelus costae from Iskenderun Bay, including length-weight relationships, growth parameters, and molecular diversity indicators. The results confirm that the population is currently in a stable condition, with positive allometric growth and relatively high haplotype diversity. However, signs of moderate nucleotide diversity and potential sensitivity to environmental stressors suggest that the population may be vulnerable to future pressures.

## ACKNOWLEDGEMENTS

I would like to thank Prof. Dr. Cemal Turan for his help in conducting the study, analyzing the data, and interpreting the results.

### Funding
The author has received no funding for this work.

### Competing Interests
Servet Ahmet Doğdu is an Academic Editor for PeerJ.

### Author Contributions
- Servet Ahmet Doğdu conceived and designed the experiments, performed the experiments, analyzed the data, prepared figures and/or tables, authored or reviewed drafts of the article, and approved the final draft.

### Animal Ethics
The following information was supplied relating to ethical approvals (i.e., approving body and any reference numbers):

Since dead fish material was used in the study, it does not require Local Ethics Committee Approval since experimental animals were not used.

## DNA Deposition

The following information was supplied regarding the deposition of DNA sequences:

The sequences are available at NCBI: PQ814215–PQ814244.

## Data Availability

The raw data is available in the Supplemental Files.

## Supplemental Information

Supplemental information for this article can be found online at http://dx.doi.org/10.7717/peerj.19594#supplemental-information.

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
