# Peer review of "Biological and genetic structure of Epinephelus costae (Steindachner, 1878) population in Iskenderun Bay, eastern Mediterranean Sea"

_PeerJ, doi:10.7717/peerj.19594_

## Round 0.1 · original submission · Major Revisions

Dear Dr. Doğdu

You can find the comments and suggestions of the expert reviewers in the attached reports. As you will see, expert reviewers have pointed out the critical errors. Consequently, a major revision is needed for your article.

I request you check and correct the manuscript based on the reports.

Sincerely

**Language Note:** The review process has identified that the English language must be improved. PeerJ can provide language editing services - please contact us at [email protected] for pricing (be sure to provide your manuscript number and title). Alternatively, you should make your own arrangements to improve the language quality and provide details in your response letter. – PeerJ Staff

Reviewer 1 ·

Basic reporting

This study provides the first comprehensive analysis of the biological (length-weight relationship, age-growth dynamics) and genetic (mitochondrial DNA diversity) characteristics of Epinephelus costae in Iskenderun Bay, Türkiye.

The manuscript is written in clear, professional English. However, certain sections (particularly the Discussion) could benefit from streamlined phrasing for improved readability. For biological analyses, the sample size (325 individuals) is sufficient for length-weight and age-growth analyses. However, the absence of age-0 individuals and limited maximum age (5 years) may restrict demographic inferences. Scale-based aging is standard, but cross-validation with otoliths or alternative methods could strengthen results. The mtDNA COI region is appropriate for population genetics, but the small subset (30/325 individuals) raises concerns about representativeness.

Experimental design

The length-weight relationship (R²=0.9941) is robust, though the b-value (3.0839; positive allometry) contrasts with prior studies (e.g., Çete 2017: 3.310). Environmental or seasonal factors (sampling limited to Jan-Apr 2024) may explain discrepancies. High haplotype diversity (0.9379) and moderate nucleotide diversity (0.0332) indicate a stable but historically bottlenecked population. Neutrality tests (Tajima’s D=-0.94, Fu’s Fs=-0.034) suggest weak expansion signals, conflicting with Achaz’s Y-test (-1.87). These inconsistencies warrant further analysis (e.g., microsatellites) to resolve demographic history.

Validity of the findings

Implications for fisheries management (e.g., stock sustainability, overfishing risks) are not sufficiently developed in the discussion section. I also recommend professional editing for clarity (e.g., complex sentences in the Discussion). In this regard, my recommendation as a reviewer regarding this study is a minor revision. Please address technical clarifications, expand fisheries management implications, and improve readability.

·

Basic reporting

Overall, this article provides detailed and valuable data on the biology and genetic structure of the Iskenderun population of Epinephelus costae. The methodologies used for assessing the length-weight relationship, age-growth characteristics, and genetic diversity are robust and clearly presented.

Experimental design

The research question regarding the biological and genetic characteristics of Epinephelus costae in Iskenderun Bay is well-defined, relevant, and effectively addresses a significant knowledge gap. Indeed, the authors explicitly state that detailed data on the length-weight relationship, growth parameters, and genetic diversity of this economically important species were previously lacking, highlighting the contribution of their study in filling this gap.

Validity of the findings

Biological and genetic analyses were methodically performed, although some additional statistical details should be included for completeness (see specific comments). Conclusions appropriately answer the original research question without overstating results. However, given the relatively small size range (16.7–43.5 cm) of the more than 300 specimens examined, it is evident that the population studied consists primarily of juveniles or subadults. This aspect should be explicitly acknowledged and discussed, particularly its implications for growth parameter estimations and management strategies. Additionally, the English language requires thorough revision to improve clarity and readability, ensuring the manuscript meets publication standards.

Additional comments

Line 20: remove "very".
Line 41: specify that E. alexandrinus is the invalid name now attributed to E. fasciatus.
Line 141: Considering the length and weight of specimens, they should all be considered juveniles, and this must be mentioned in the discussion.
Line 146-147: you should perform a t-test to state this, if the b is statistically over 3 for a positive allometric growth.
Line 212-215: Can you verify if Samir et al. and Magnusson et al. reported exactly the same "b" value for E. aeneus?

·

Basic reporting

This article lacks clarity in conveying the problem statement and the solution offered to solve the problem. The author needs to focus on issues relevant to the target species and align with appropriate literature. Even though the author has presented the latest literature on the target species, he has not been able to present a comprehensive writing between his thoughts and study results to the literature.

Experimental design

The author did not correctly write the research question, so it could not identify the knowledge gap and how this study fills the gap. Statements made by the author tend to be personal assumptions without being supported by data and facts. Sample collection in the field is not explained in detail, raising the question of whether the samples collected represent the characteristics of the target species population. Some analyses were conducted using Generative AI technology without author evaluation, resulting in misconceptions and unusual graphs.

Validity of the findings

The provided data cannot explain the representativeness of the coverage area that the author claims in the article title. The author needs to explain the reasons for choosing sampling sites to ensure that the collected data sufficiently represents the population. Conclusions are also less relevant to the Results and Discussion section and tend to overclaim.

Additional comments

Please see the additional file that I attached for a detailed review.

---

## Round 0.2 · accepted · Accept

Dear Dr. Doğdu

I thank you for making the corrections and changes requested by the reviewers. I read and checked your valuable article carefully and am happy to inform you that the article has been accepted for publication in PeerJ.
Sincerely yours,